# The Study on Timolol and Its Potential Phototoxicity Using Chemical, In Silico and In Vitro Methods

**DOI:** 10.3390/ph17010098

**Published:** 2024-01-11

**Authors:** Karolina Lejwoda, Anna Gumieniczek, Agata Filip, Beata Naumczuk

**Affiliations:** 1Department of Medicinal Chemistry, Medical University of Lublin, 20-090 Lublin, Poland; k.lejwoda94@gmail.com; 2Department of Cancer Genetics, Cytogenetics Laboratory, Medical University of Lublin, 20-080 Lublin, Poland; agata.filip@umlub.pl; 3Institute of Organic Chemistry Polish Academy of Sciences, Kasprzaka 44/52, 01-224 Warsaw, Poland; beata.naumczuk@icho.edu.pl

**Keywords:** timolol, photodegradation, phototoxicity, LC-UV and UPLC-HRMS, ROS generation, in vitro tests, fibroblasts

## Abstract

Timolol (TIM) is a non-selective ß-adrenergic receptor antagonist used orally for the treatment of hypertension and heart attacks, and topically for treating glaucoma; lately, it has also been used in some specific dermatological problems. In the present study, its photodegradation and potential risk of phototoxicity were examined using chemical, in silico and in vitro methods. The UV/VIS irradiated solutions of TIM at pH 1–13 were subjected to LC-UV and UPLC-HRMS/MS analyses showing pseudo first-order kinetics of degradation and several degradation products. The structures of these photodegradants were elucidated by fragmentation path analysis based on high resolution (HR) fragmentation mass spectra, and then used for toxicity evaluation using OSIRIS Property Explorer and Toxtree. Potential risk of phototoxicity was also studied using chemical tests for detecting ROS under UV/VIS irradiation and in vitro tests on BALB/c 3T3 mouse fibroblasts (MTT, NRU and Live/Dead tests). TIM was shown to be potentially phototoxic because of its UV/VIS absorptive properties and generation ROS during irradiation. As was observed in the MTT and NRU tests, the co-treatment of fibroblasts with TIM and UV/VIS light inhibited cell viability, especially when concentrations of the drug were higher than 50 µg/mL.

## 1. Introduction

Knowledge on intrinsic stability of drugs under their exposure to light is crucial for pharmaceutical industry and clinicians for various reasons. On the one hand, it is important from the point of effective drug manufacturing. On the other hand, potential side effects due to drug degradation could be observed in patients, especially when the drugs are topically applied or being accumulated in the skin [1].

Timolol (TIM) is a non-selective ß-adrenergic receptor antagonist used as a maleate salt in ophthalmic solutions for treating ocular hypertension and open-angle glaucoma. It could also be orally used for the treatment of hypertension and heart attacks as well as in prevention of migraine headaches [2,3]. Chemically, TIM is 1-(*tert*-butylamino)-3-[(4-morpholin-4-yl-1,2,5-thiadiazol-3-yl)oxy]propan-2-ol applied as a pure (*S*) enantiomer, which is around 50 times more active than its (*R*) enantiomer. On the other hand, this (*R*) enantiomer is described as showing less hazardous side effects. What is more, it is reported that the (*R*) enantiomer could show beneficial effects on intraocular pressure without the undesirable side-effect of bronchial constriction caused by non-selective action of (*S*)-TIM [4,5] (Figure 1).

In addition to typical uses of β-blockers such as hypertension, arrhythmias and heart failure, TIM has gained a big interest amongst dermatologists based on its demonstrated effects in specific dermatological disorders, i.e., hemangiomas, Kaposi sarcoma, melanoma, pyogenic granuloma and pemphigus, wound healing and prevention of radiation-induced dermatitis. On the other hand, β-blockers could cause some undesired skin reactions in many patients, both after systemic and topical applications [6]. What is more, many incidents of photosensitization have been reported for them as well as some photoinduced cutaneous reactions [7]. For these reasons, it is very important to procure more data about photoreactivity of this important grup of drugs, including TIM.

The studies confirming the drug stability, as well as experiments on accelerated photodegradation, are important parts of drug stability testing, according to ICH Q1B guidelines [8]. First, it is necessary to determine the degree and rate of photodecomposition over time. Secondly, it is necessary to detect all degradants and define their structures to assess their potential toxicity for patients and risk due to their presence in respective pharmaceuticals. Recently, chemical tests, in silico methods and in vitro models have been recommended to examine potential toxic effects caused by photodegradants present in the marketed formulations [1,9].

Taking into account the ability of TIM to absorb UV/VIS light above 300 nm, the assessment the risk of damage to skin is of great importance in view of its topical applications as well as its skin adverse effects observed after systemic administration [10]. Therefore, we performed a detailed investigation on photostability of TIM using quantitative LC-UV and qualitative UPLC-HRMS/MS analyses; on potential phototoxicity using chemical tests for generation of reactive oxygen species (ROS); and finally, in silico and in vitro methods. The LC-UV experiments were carried out for quantitative measurements of TIM under exposure of UV/VIS light in solutions of pH 1–13, and finally for kinetic measurements. The next steps were focused on identification of degradation products using UPLC-HRMS/MS experiments, and predicting their toxicity using OSIRIS Property Explorer and Toxtree [11,12]. In the next step, the possibility of generation of singlet oxygen (^1^O_2_) and superoxide radical anion (O_2_**^−^**) under exposure of TIM to UV/VIS irradiation was assessed. Finally, phototoxic potential of TIM and its degradation products were examined using in vitro methods on the cultured mouse fibroblasts.

It is worth noting that the literature on photosensitivity of TIM and its potential risk of phototoxicity in patients, and the risk of its excessive degradation in marketed formulations, is not very extensive. Only few papers on degradation of TIM including photolytic conditions were found [3,13,14,15,16]. Between others, these findings suggest that photodegradation is a significant pathway that could be used for elimination of TIM and other β-blockers from natural waters [13,14,15].

## 2. Results and Discussion

### 2.1. UV/VIS Measurements

Photosensitive drug substance could be excited by absorption of light energy, and then can react with other substances through energy transfer and/or radical reactions, with the possibility of photoinduced toxicity. Thus, an UV/VIS analysis is recommended by OECD guidelines as the initial step of testing drug phototoxicity [17]. Therefore, in the present study, the UV/VIS analysis was carried out over the range of 200–700 nm to clarify the photoreactivity of TIM. According to the spectral results obtained, considerable UV absorption was observed, especially in the range of 200–400 nm. In addition, TIM showed a large peak with a maximum at ca. 300 nm, with the molar extinction coefficient (MEC) higher than 1000 M^−1^ cm^−1^. Based on of these results, TIM was recognized as being potentially photoreactive [17,18,19]. What is more, after irradiation at different pH, the UV spectra of TIM showed visible differences in relation to non-irradiated ones, as far as the maximum absorbance peak of the shape of the absorption curve were concerned. Figure 2 presents the changes in the sample of TIM in the buffer of pH 7.4 after UV/VIS irradiation with a dose 2015 kJ/m^2^.

### 2.2. Validation Parameters of LC-UV Method

Selectivity, linearity, limit of detection (LOD) and limit of quantitation (LOQ), precision and accuracy were evaluated as validation parameters for our LC-UV method, according to the respective ICH guidelines [20]. The obtained results from the validation study are shown in Table 1.

### 2.3. Photodegradation of TIM and Kinetic Measurements

Photodegradation of TIM was estimated with the LC-UV method described above, after exposing the drug to UV/VIS light for 7, 14, 21, 28 and 35 h with the energies 18,902, 37,804, 56,706, 75,608 and 94,510 kJ/m^2^, respectively. As a result, degradation of 82.38–98.13% was observed after the maximal dose of irradiation, depending on pH value. In addition, the obtained LC-UV chromatograms show the peak of non-degraded TIM (t_R_ = 2.283 min) decreasing with a corresponding presence of additive peaks of degradation products (Figure 3).

It is well known that the pH of the environment could promote or inhibit photolysis, and drug stability could change depending on pH value. This is because the ionized or non-ionized forms of drug molecules could vary in stability and may undergo specific acid–base catalysis in aqueous solutions [21]. TIM presents the basic pKa of 9.2 due to its *tert*-butylamino group [22], and is shown to be more labile in its ionized form in acidic medium. On the contrary, it shows lower photodegradation in neutral and alkaline conditions at pH 7–13 (Figure 4).

Based on our results, stronger correlations with R^2^ values above 0.9 were obtained for the plots of logarithms of concentration of non-degraded TIM than for the plots of concentration of non-degraded TIM, pointing at the pseudo first-order kinetics. Based on respective linear equations, further kinetic parameters were calculated, i.e., degradation rate constant (k), degradation time of 10% substance (t_0.1_), and degradation time of 50% substance, confirming the lowest stability of TIM in acidic conditions (Table 2). 

In the literature, no results on degradation of TIM at different pH have been published so far. However, in the studies of Piram et al. [13,14], photostability of ten ß-blockers, including TIM, was studied in environmental waters. As a result, TIM showed pseudo-first order kinetic,. and its t_0.5_ values were calculated in the range 3–5 h. TIM was also described as hardly degraded under simulated sunlight with a t_0.5_ of 33 h in pure water [16]. However, in our study, sensitivity of TIM to the light was showed over the entire pH range with the extrema at pH 1–4. It could be an essential question for the manufacturing of new formulations containing TIM and protecting them during their presence on the market. The pH value could be the stability-controlling factor for TIM formulation products, especially for ocular ones, but also for other topical forms [1].

### 2.4. LC-MS Analyses Results

In the literature, one photodegradation product of TIM was identified and characterized by LC-ESI/MS/MS after forced degradation in a solid state. The detected product was described as the protonated 4-morpholino-1,2,5-thiadiazol-3-ol with m/z 188, showing that TIM can degrade by elimination of C_7_H_15_NO [3]. In the present study, identification of photodegradants of TIM was performed on a basis of UPLC-MS analysis and the proposed structures were elucidated by fragmentation path analysis based on HR fragmentation mass spectra. By analyzing the LC-UV-HRMS results of TIM after UV/VIS irradiation at pH 4, it was possible to identify four degradation products, PD1–PD4 (Figure 5). 

In a respective UV chromatogram (Figure 5), the peak corresponding to TIM is present at t_R_ = 3.03 min. The full scan mass spectra of TIM were obtained, in which the protonated molecule at m/z 317 was detected. Then, the MS/MS spectrum showed the product ion at m/z 261 due to elimination of *tert*-butyl group that formed subsequent ions at m/z 74 and 244 (the loss of NH_3_). The last could dissociate to the ion at m/z 188 via breakage of the ether bond and further to subsequent ions at m/z 113 and 144. A protonated ion of TIM could also follow other fragmentation pathways, the first leading to an ion at m/z 130, the second with an ion at m/z 57 and the third forming the aforementioned ion at m/z 188. The proposed fragmentation pattern for TIM is presented in Figure 6.

The full scan mass spectra of the peak with t_R_ = 2.10 min were obtained in which the protonated molecule at m/z 247 was detected. The MS/MS spectrum of this degradation product (PD1) shows the product ion at m/z 191 formed after elimination of *tert*-butyl group and NH_3_, producing an ion at m/z 174. This ion and the molecular ion at m/z 247 can formed the ion at m/z 118 after the ether bond breakdown. Thus, this fragmentation pattern (Figure 7) is similar to the fragmentation pattern of parent TIM and suggests that both compounds differ structurally only in a part of 1,2,5-thiadiazole ring. The analysis of this fragmentation pattern, accurate mass measurements and elemental composition indicate the structure of PD1 as 3-[(4-amino-1,2,5-thiadiazol-3-yl)oxy]-*N*-*tert*-butyl-2-hydroxypropan-1-aminium.

A next degradant from experiments at pH 4, i.e., PD2 with m/z 291, was formed by opening the morpholine ring of the parent structure of TIM. Subsequent fragmentation steps are similar to TIM and PD1, i.e., elimination of *tert*-butyl group (m/z 235) and NH_3_ (m/z 218), and also breaking the ether bond (m/z 162), which leads to the proposed structure of *N*-*tert*-butyl-2-hydroxy-3-({4-[(2-hydroxyethyl)amino]-1,2,5-thiadiazol-3-yl}oxy)propan-1-aminium (Figure 8).

As far as PD3 (m/z 331) is concerned, hydroxylation of the morpholine ring together with its dehydrogenation is proposed. The MS/MS spectrum of the molecular ion at m/z 331 shows two product ions, which are similar to the fragmentation patterns described above, i.e., elimination of *tert*-butyl group (m/z 275) and breakage the ether bond (ion m/z 202). Moreover, the ion at m/z 275 after the loss of H_2_O gives the m/z 257 product ion, confirming the presence of a hydroxyl group in the structure of this degradation product. Analysis of the fragmentation pattern (Figure 9) from the MS/MS spectrum, accurate mass measurements and elemental composition indicate the proposed structure of PD3 as *N*-*tert*-butyl-2-hydroxy-3-{[4-(2-hydroxy-2,3-dihydro-4H-1,4-oxazin-4-yl)-1,2,5-thiadiazol-3-yl]oxy}propan-1-aminium.

Based on similarities in fragmentation patterns to TIM and its other degradants, accurate mass measurements and elemental composition, PD4 (m/z 315) appears to be the product of dehydrogenation of morpholine ring and was identified as *N*-*tert*-butyl-3-{[4-(2,3-dihydro-4H-1,4-oxazin-4-yl)-1,2,5-thiadiazol-3-yl]oxy}-2-hydroxypropan-1-aminium (Figure 10).

By analyzing the LC-UV-HRMS results of TIM after UV/VIS irradiation at pH 13, it was possible to identify one degradation product, PD5. In a respective UV chromatogram (Figure 11), two main peaks are present. The peak with t_R_ = 3.03 min and m/z 317 in an amount of approximately 73.3% belongs to TIM, and the second one with t_R_ = 2.17 min and m/z 204 in an amount of approximately 22% belongs to PD5.

At pH 13, photodegradation of TIM occurrs through oxidation of the S atom in the 1,2,5-thiadiazole ring, leading to respective dioxide, and by removal of *tert*-butylaminopropoxy side chain from the parent drug structure. The mass spectrum for the peak with t_R_ = 2.17 min shows the protonated ion [M+H]^+^ with m/z 204. The analysis of the fragmentation pattern from the MS/MS spectrum suggests an absence of product ions due to elimination of the *tert*-butyl group, NH_3_ and the ether bond breakage, which confirm the lack of respective side chain in 1,2,5-thiadiazole ring. However, the product ion at m/z 140 is present due to elimination of sulfur dioxide, meaning that PD5 contains the SO_2_ group. Thus, the proposed structure of PD5 is 4-(1,1-dioxo-1H-1γλ^6^,2,5-thiadiazol-3-yl)morpholin-4-ium (Figure 12).

Taken together, the degradation of TIM could be the result of different processes at pH 4 and pH 13. On the other hand, decomposition generally occurred at the morpholine ring or *tert*-butylaminopropoxy side chain in the parent structure of TIM. It is also worth noting that at least four of the above degradation products PD1 and PD3–PD5 have not been reported so far [3]. At the same time, the observed degradation was proven to be light-induced because any of these degradants had not been detected after thermolytic conditions at identical pH. The summative scheme for TIM degradation at different pH is proposed in Figure 13. 

Finally, OSIRIS Property Explorer (https://www.organic-chemistry.org/prog/peo/tox.html, accessed on 9 December 2023) and Toxtree 3.1.0 software [11,12] were used to predict mutagenic, tumorigenic, irritant and reproductive effects of PD1-PD5. As a result, OSIRIS did not show toxicity risk for the mentioned degradants. When Toxtree was used, the class III of toxicity that is characteristic of the compounds containing functional groups associated with enhanced toxicity was obtained for PD1-PD4. This is because of the presence of *N*-methyl *tert*-butylamine, similar to the parent structure of TIM. On the other hand, the structure similar to PD2 was detected after thermolytic stress of TIM in the acidic medium and then estimated as having the risk of toxicity using FDA Carcinogenecity Male Rat Single vs. Mult v3.1 and skin Sensitization NEG vs SENS v6.1 by TOPKAT approach in Accelrys Discovery Studio 2.5 (Dassault Systemes BIOVIA, San Diego, CA, USA) [3].

### 2.5. ROS Generation

After examining the MEC value for TIM that was showed >10,000 and finding its huge degradation under UV/VIS irradiation, it seemed interesting to explore the possibility of ROS generation. Based on the official recommendations [17,18,19], the potential phototoxicity of the drugs could be examined by the detection of singlet oxygen (^1^O_2_) and superoxide radical anion (O_2_**^−^**) due to type 2 and type 1 photochemical reactions. Generally, the drug is being estimated as photoreactive when ^1^O_2_ values are ≥25 and O_2_**^−^** values are ≥70, when ^1^O_2_ values are less than 25 but O_2_**^−^** values are ≥70, and when ^1^O_2_ values are ≥25 and O_2_**^−^** values are less than 70. When ^1^O_2_ values are lower than 25 and O_2_**^−^** values are between 20 and 70, the drug could be clasified as weakly photoreactive [17]. According to these criteria, TIM could be estimated as a phototoxic drug generating ROS under UV/VIS irradiation, because ^1^O_2_ is higher than 25 while energy 234 kJ/m^2^ was used (Table 3). Thus, further experiments concerning its potential phototoxicity should be performed, as well as how it can be increasingly used in topical applications where the risk of photoreactivity is higher. What is more, such photoreactivity could be highly relevant to cause some adverse effects in patients. It is well known that toxic responses are elicited through topically administered photoreactive drugs after the exposure of patients to environmental light [1,2,23]. What is more, the ROS test may provide more information than other tests, e.g., the NRU (Neutral Red Uptake) phototoxicity test. For example, it is confirmed that tests detecting ROS can better predict the risk of photoallergy [17]. 

### 2.6. In Vitro Tests

MTT in vitro Toxicology Assay (3-[4,5-dimethylthiazol-2-yl]-2,5 diphenyl tetrazolium bromide assay) determines metabolic reduction of this specific reagent in live cells, leading to the formation of purple formazan crystals proportional to cell viability. First, the percentage of cell viability of each group co-treated with TIM at concentrations from 0 to 1000 µg/mL and UV/VIS with a dose 5 J/cm^2^ was calculated in relation to the non-irradiated cells incubated with similar TIM concentrations. Our final results are expressed as the viability of cells irradiated in the presence of TIM at concentrations of 5–1000 µg/mL in comparison to the sample exposed to irradiation without TIM (0 µg/mL). Under such co-treatment, viability of fibroblasts irradiated in the presence of TIM was in the range of 78.21–50.22%, with increasing TIM concentration from 5 to 1000 µg/mL. A statistically significant (at *p* < 0.005) decrease in cell viability was observed at TIM concentrations of 10 µg/mL, while the concentration at which the cell viability was reduced by approximately 50% is 500 µg/mL (Figure 14a).

As far as the NRU test is concerned, NR dye accumulates in lysosomes of live cells, proportionally to cell viability. Similar to our MTT assay, the results are expressed as the percentage of cell viability of each group treated with TIM at concentrations from 0 to 1000 µg/mL and irradiated with a dose of 5 J/cm^2^, relative to respective non-irradiated samples. Then, our final results are expressed as viability of cells irradiated in the presence of TIM at concentrations of 5–1000 µg/mL in comparison to the sample exposed to irradiation without TIM (0 µg/mL). Under this co-treatment, a visible drop in cell viability (75.44–42.32%) was observed. A statistically significant impact of TIM was observed at concentration 25 µg/mL at *p* < 0.02, and at concentration 50 µg/mL at *p* < 0.05. At the same time, irradiated fibroblasts with the concentration of TIM equal to 500 µg/mL showed the reduced cell viability by approximately 50% (Figure 14b). 

Using the Live/Dead Test, simultaneous determination of live and dead cells was achieved using two reagents that reflect the intracellular esterase activity and membrane integrity [24]. In live cells, calcein acetoxymethyl ester (CA-AM) is converted by esterases, yielding cytoplasmic green fluorescence. On the other hand, in membrane-affected dead cells, ethidium homodimer-1 (EthD-1) binds to nucleic acids with intensive red fluorescence. After irradiation with the dose of light of 5 J/cm^2^, it was showed that TIM at a concentration of 1000 µg/mL reduced the total cell population by about 10%, similar to the probe irradiated without TIM, in comparison with respective non-irradiated samples (Figure 15).

Generally, the higher the dose of TIM, the less viability of the fibroblasts was observed with our MTT and NRU tests. However, this trend was not confirmed in the Live/Dead fluorescence test, showing that this last test was less sensitive. Thus, it could be concluded that more than one test should be performed to examine cell viability in order to avoid overestimation or underestimation of these phototoxic effects. 

## 3. Materials and Methods

### 3.1. Materials for Chemical Tests

Pharmaceutical grade standards of timolol maleate (TIM), papaverine hydrochloride, quinine hydrochloride, benzocaine, imidazole, p-nitrosodimethylaniline (RNO), nitroblue tetrazolium chloride dye (NBT), sodium 1-heptanesulfonate, neutral red dye (NR), dimethyl sulfoxide (DMSO), ethanol, methanol from Merck (Darmstad, Germany), acetonitrile, formic acid and water for LC/MS from J.T. Baker (Center Valley, PA, USA) were used. Acetic acid, hydrochloric acid, phosphoric acid, sodium acetate, chloride, tetraborate, hydrogen phosphate and dihydrogen phosphate, kalium hydroxide and kalium dihydrogen phosphate were purchased from POCh (Gliwice, Poland). The buffers of pH 4, 7 and 10 were prepared as was described in European Pharmacopoeia [25]. Phosphate buffer of pH 7.4 (20 mM) was prepared according to respective literature [17].

### 3.2. LC-UV Method

A chromatograph with a model 515 pump, a Rheodyne 20 µL injector and a model UV 2487 DAD working at 280 nm controlled by Empower 3 software, all from Waters UK Sales (Elstree, UK), were used. Analysis was carried out on a LiChrospher^®^RP8 column (125 × 4.0 mm, 5 µm) from Merck, with a mixture of 140 mL of water, 60 mL of acetonitrile, 1 mL acetic acid and 0.22 g sodium heptanesulfonate as the mobile phase with the flow rate of 2 mL/min.

Selectivity of the method was examined by determination of TIM in the presence of its degradation products. For calibration, working solutions in the concentration range of 0.01–0.10 mg/mL were prepared by dispensing 0.1–1.0 mL volumes from the stock solution of TIM (1 mg/mL), adding 0.5 mL volumes of the stock solution of i.s. (papaverine hydrochloride, 1 mg/mL) and diluting to 10 mL. Calibration equations were obtained by plotting the ratio of peak areas (TIM versus i.s.) against corresponding concentrations of TIM. The LOD and LOQ were determined using the SD values of the intercept and slope from the regression equations. Injections of the next working solutions of TIM at low (15 µg/mL), medium (50 µg/mL) and high (90 µg/mL) concentrations were conducted and analyzed to verify accuracy and repeatability of the method. Percentage recovery of TIM was used as accuracy, while the within-day and between-day estimations were used as precision indexes.

### 3.3. UPLC-HRMS/MS Analysis

An ACQUITY UPLC I-Class chromatograph coupled with a Synapt G2-S HDMS mass spectrometer equipped with an electrospray ion source and a q-TOF type mass analyzer from Waters were used for UPLC-MS/MS analysis. Separations were carried out using an Acquity UPLC BEH C18 column (2.1 × 100 mm, 1.7 µm) from Waters with the following gradient conditions: from 5% to 100% of eluent B (acetonitrile) over 10 min, at a flow rate of 0.3 mL/min, while eluent A was formic acid in water (0.1%, *v*/*v*). The UV chromatograms were recorded at 280 nm.

High Resolution spectra in a positive mode were obtained with capillary voltage set to 3.0 kV. The desolvation gas flow was 600 L/h, while the temperature was 250 °C. The sampling cone voltage 40 V, the source offset 80 V and the source temperature 100 °C were applied. The data were obtained in a scan mode ranging from 100 to 1500 m/z. Leucine-Enkephaline solution was used as the lock-spray reference material. MassLynx V4.1 software package from Waters was used for processing the recorded data.

### 3.4. Preliminary UV/VIS Analysis

Working solutions were prepared by dissolving TIM in methanol (1 mg/mL) and then diluting to a final concentration of 10 µg/mL with 20 mM phosphate buffer of pH 7.4. A UV/VIS double beam spectrophotometer (CE 6600 from Cecil Instruments Ltd., Milton, UK) and quartz cells with 1 cm path length were used for recording the absorptive spectra of irradiated and non-irradiated samples in the range 200–700 nm. The MEC values were calculated from the absorbance values recorded at the maximum wavelengths.

### 3.5. UV/VIS Light Simulator

All UV/VIS irradiations were performed in the range 300–800 nm using a Suntest CPS Plus chamber from Atlas (Linsengericht, Germany) equipped with the temperature control unit set at 35 °C. 

### 3.6. Photodegradation of TIM in Solutions

Volumes of 1 mL of the stock solution of TIM (1 mg/mL) were dispensed to small quartz glass-stoppered dishes and diluted with 1 mL of respective solvent, i.e., 0.1 M HCl (pH 1), buffers of pH 4, 7, 10 and 0.1 M NaOH (pH 13). The samples were exposed to UV/VIS irradiation for 7, 14, 21, 28, and 35 h with energies 18,902, 37,804, 56,706, 75,608 and 94,510 kJ/m^2^, respectively. After irradiation, samples were neutralized if necessary and diluted with methanol, and analyzed quantitatively by our LC-UV method. The concentrations of non-degraded TIM for each time of irradiation were calculated from the calibration equation and the starting concentration of TIM.

### 3.7. Kinetics of Photodegradation

Concentrations of non-degraded TIM remaining after each round of irradiation or logarithms of these concentrations were plotted against time of degradation to obtain linear equations y = ax + b and determination coefficients R^2^, and finally to determine the reaction order. Then, further kinetic parameters, i.e., degradation rate constant (k), degradation time of 10% substance (t_0.1_), and degradation time of 50% substance (t_0.5_), were calculated.

### 3.8. ROS Assays [17]

For two ROS assays, the 20 mM stock solution of TIM in DMSO was prepared and diluted with phosphate buffer of pH 7.4 to obtain the 1 mM working solution. Similar solutions of quinine and benzocaine were prepared (positive and negative controls, respectively). Also, 0.2 mM solutions of RNO and imidazole, and 0.4 mM solution of NBT, were prepared in phosphate buffer of pH 7.4.

The assay of singlet oxygen ^1^O_2_ was based on bleaching RNO in the presence of imidazole as a selective acceptor of ^1^O_2_. The samples of TIM (20 µL) were mixed with RNO (250 µL) and imidazole (250 µL), and diluted to 1 mL with phosphate buffer of pH 7.4. After throughly mixing, the absorbance of each sample was measured at 440 nm. Then, the samples were irradiated under UV/VIS with energies described below, and a final decrease of absorbance was measured at 440 nm. The assay of superoxide radical anion O_2_**^−^** was based on monitoring the reduction of NBT. The samples of TIM (20 µL) were mixed with NBT (125 µL) and diluted to 1 mL with a phosphate buffer of pH 7.4. After throughly mixing, the absorbance of each sample was measured at 560 nm. Then, the samples were irradiated under UV/VIS with respective energies, and an increase of absorbance was measured at 560 nm. 

In both tests, samples of TIM were irradiated for 1, 5, 15, 30, 45 min and for 1 h (49, 234, 675, 1350, 2025, 2700 kJ/m^2^). The selected energies comprised the range 5–20 J/cm^2^ recommended in the literature [17,18,19], as well as some energies above. The final concentration of TIM in the irradiated samples was 20 µM. All measurements were repeated three times for each sample using an UV/VIS double beam spectrophotometer from Cecil, describe above.

### 3.9. Cell Cultures

In vitro studies were performed on mice fibroblasts (BALB/c 3T3) from ATCC (Manassas, VA, USA). The cells were pre-incubated at 5% CO_2_ humidity and 37 °C in the appropriate grow medium, i.e., Dublecco’s Modified Eagles Medium (DMEM) with L-glutamine, bovine calf serum and Antibiotic-Antimycotic Solution (10,000 units of penicillin, 10 mg of streptomycin and 25 μg of amphotericin B per 1 mL) (Sigma-Aldrich, St. Louis, MO, USA). All cells used in the present experiments were from the passages from 5 to 10. They were seeded into the 96-well microplates and on the glass coverslips. For further steps, phosphate buffered saline (PBS), DMEM-PBS, DMEM without phenol red and Hank’s Balanced Salt Solution (HBSS) from Sigma were used. 

### 3.10. MTT Test [26]

The 96-well plates with seeded fibroblasts and 100 µL of DMEM were incubated for 24 h at 37 °C. Then, the cells were washed with PBS and the culture medium was replaced with 100 µL of TIM at eight concentrations, ranging from 5 to 1000 µg/mL in DMEM without phenol red. After incubation for 4 h at 37 °C, the UV/VIS (300–800 nm) irradiation was performed through the lids of the plates with a dose of 5 J/cm^2^. At the same time, the non-irradiated cells were kept in dark conditions. After this, all cells were washed with PBS and re-incubated for the next 20 h with 100 µL of DMEM. After washing with PBS and diluting with 100 µL of DMEM without phenol red, 10 µL of MTT solution in PBS (5 mg/mL) from MTT in vitro Toxicology Assay kit Tox-1KT (Merck) was added to each well. After 4 h incubation, 100 µL of respective stop solution from the above MTT kit was added instead of the medium. After shaking thoroughly, the absorbance was assayed at 560 nm using a microplate reader Tecan Sunrise controlled by the Magellan v7.1 software from Tecan Trading SA (Mannedorf, Switzerland), utilizing the respective controls. All experiments were carried out with at least six parallel wells.

### 3.11. NRU Test [27]

The 96-well plates with seeded fibroblasts and 100 µL of DMEM were incubated for 24 h. Then, they were washed with 150 µL of HBSS and the culture medium was replaced with 100 µL of TIM at eight concentrations, ranging from 5 to 1000 µg/mL in PBS. After incubation for 4 h at 37 °C, the UV/VIS (300–800 nm) irradiation was performed through the lids of the plates with a dose of 5 J/cm^2^. At the same time, the non-irradiated cells were kept in dark conditions. After this, all cells were washed with 150 µL of HBSS and then re-incubated for the next 20 h with 100 µL of DMEM. After washing with 150 µL of HBSS, 100 µL of NR in DMEM (50 µg/mL) was added to each well. After 4 h incubation and washing cells with the buffer, 150 µL of the mixture containing ethanol, water and acetic acid (50 + 49 + 1) was added instead of the medium. After shaking thoroughly, the absorbance of each wll was measured at 540 nm with a microplate reader described above, utilizing respective controls. All experiments were carried out with at least six parallel wells.

### 3.12. Live/Dead Test [24]

Cells were seeded on the glass coverslips, which were then placed in small Petri dishes and cultured in DMEM (3 mL) for 48 h. Then, 2.5 mL of the medium was replaced with 0.5 mL of TIM solution in PBS to gain a concentration of 1000 µg/mL and cover the cells exactly. After incubation for 2 h at 37 °C, the UV/VIS (300–800 nm) irradiation was performed through the lids of the Petri dishes with a dose of 5 J/cm^2^. At the same time, the non-irradiated cells were kept in dark conditions. After this, all cells were washed with PBS and then re-incubated for the next 20 h with 500 µL of the grow medium. After removing the medium and washing with PBS, 500 µL of Live/Dead reagent, i.e., the mixture of calcein AM (CA-AM) and ethidium homodimer-1 (EthD-1) from Live/Dead Viability/Cytotoxicity kit from Thermo Fisher Scientific (Waltham, MA, USA), was added to each dish. CA-AM dye penetrates the live cell membranes. Upon entering the cells, intracellular esterases cleave the acetoxymethyl (AM) group from CA, yielding it the impermeable membrane. At the same time, apoptotic and dead cells in which the cell membranes are damaged do not retain CA. In turn, EthD-1 is a stain with high-affinity to nucleic acid that enters cells with damaged membranes but does not have the ability to cross the intact membrane of living cells. After 45 min of incubation, the cells were analyzed under an upright fluorescence microscope Nikon ECLIPSE Ni equipped with Nikon Plan Apo 100 × 1.45 objective and filters for FITC (excitation at 495 nm, emission at 515 nm) and Texas Red (excitation at 595 nm, emission at 615 nm) to visualize green (live) and red (dead) cells. Five images per dish were acquired at 100× magnification by means of GENESIS v7.0 imaging software from Applied Spectral Imaging, Carlsbad, CA, USA. The percent viability was calculated for each field based on the number of live and dead cells.

### 3.13. Statistical Analysis

LC-UV and ROS assay results were expressed as mean ± SD from at least three separate measurements. The results from in vitro tests are expressed as mean ± SEM from at least four experiments. Statistical analysis was performed using Statistica 13 (Tibco Software, Pale Alto, CA, USA). For MTT and NRU tests, differences among the groups that were irradiated with different TIM concentrations (0–1000 µg/mL) were assessed using one-way ANOVA followed by Tukey test, while *p* ≤ 0.05 was determined to indicate a significant difference. 

## 4. Conclusions

The present study was performed considering the frequency of use of TIM in therapy and its increasingly widespread use as ocular drops and other topical formulations. The lack of previous reports concerning its photodegradation and identification of respective degradation products was also important. We present the summative data for the degradation of TIM in solutions of different pH, its kinetics of degradation as well as identification of its degradation products. These results show that TIM is sensitive to UV/VIS light (300–800 nm) in a wide pH range 1–13, yielding at least five photodegradation products PD1-PD5. It is worth noting that four of these, i.e., PD1 and PD3–PD5 have not been described in the literature so far. When OSIRIS Property Explorer was used, none of these degradation products were shown to be mutagenic, tumorigenic, irritant or affecting reproduction. However, Toxtree in silico analysis indicated functional groups associated with enhanced toxicity in the structures PD1-PD4. In addition, TIM was shown to be able to generate ROS under irradiation, and reduce viability of the cultured fibroblasts at concentration of 50 µg/mL. As a consequence, our results provide valuable information on possible degradation processes during manufacturing or storage of respective formulations of TIM. Thus, they can be used to improve the quality of new formulations of TIM by optimal selection of environment, excipients and packing, especially when new topical formulations of TIM are to be designed. On the other hand, the results presented here can be used in further tests on the risk associated with phototoxicity of TIM and other ß-blockers in patients. 

Some limitations of the present study concern the reliability of the methods used for toxicity estimation, and sometimes possibility of its overestimation or underestimation. However, the main limitations of such studies include the inability to assess the impact of metabolic modifications of drugs in patients, as well as their individual modifications due to genetic differences. On the other hand, such experiments enrich the current knowledge on drug phototoxicity and help professionals both stay informed and better educate their patients in terms of protecting themselves and their medications against the light impact.

## Figures and Tables

**Figure 1 pharmaceuticals-17-00098-f001:**
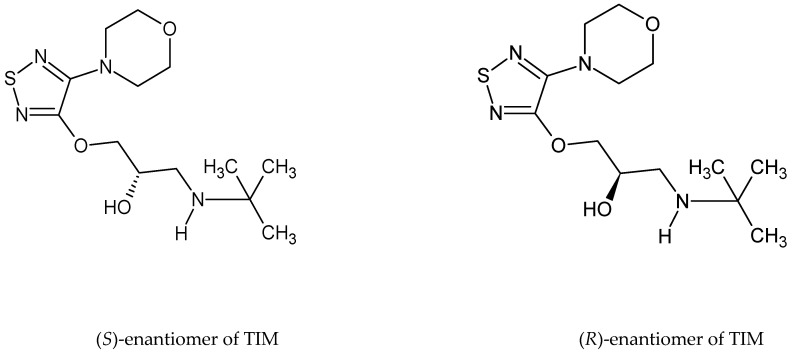
Chemical structures of timolol (TIM) enantiomers.

**Figure 2 pharmaceuticals-17-00098-f002:**
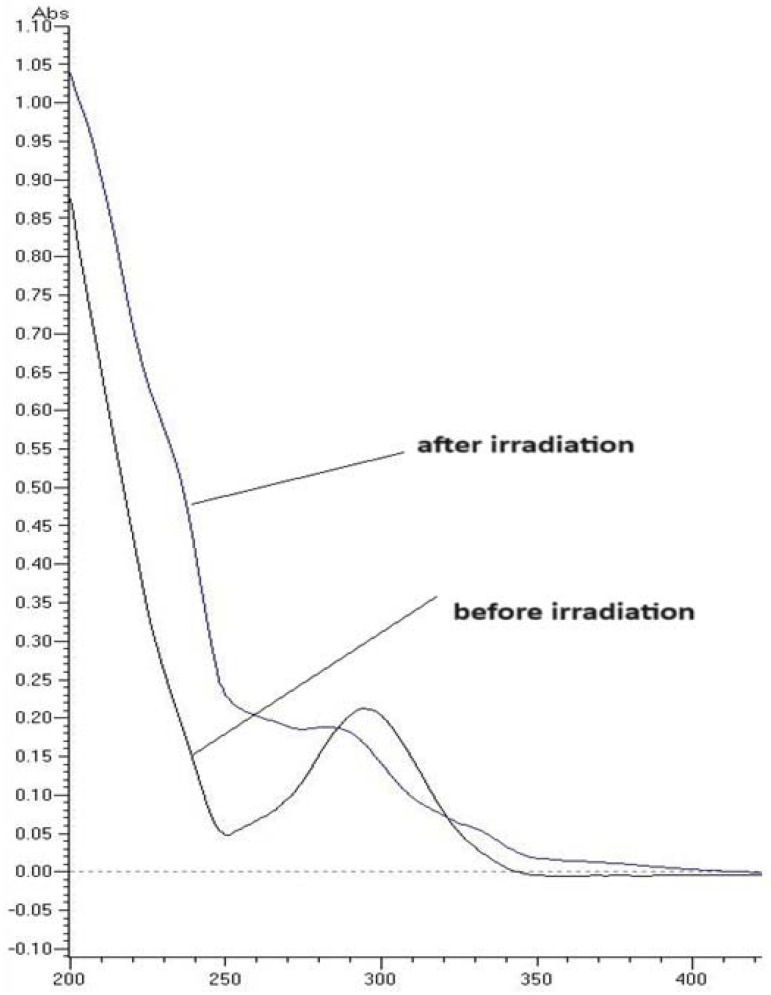
Representative absorbance spectra of timolol (TIM) (10 µg/mL) in the buffer of pH 7.4 before UV/VIS irradiation (black line) and after exposure to UV/VIS irradiation with energy 2015 kJ/m^2^ (purple line).

**Figure 3 pharmaceuticals-17-00098-f003:**
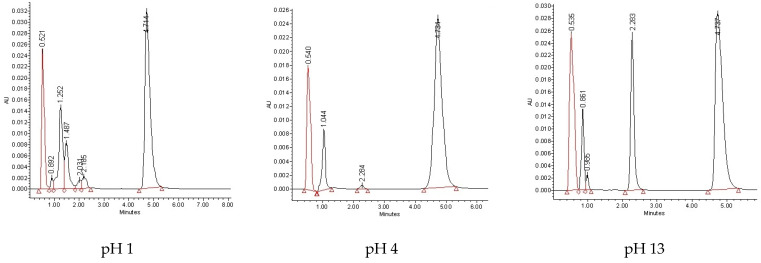
Representative chromatograms of timolol (TIM) with retention time (t_R_) = 2.283 min after UV/VIS irradiation (94.510 kJ/m^2^) at different pH in the presence of internal standard (i.s.) with t_R_ = 4.714 min and photodegradation products.

**Figure 4 pharmaceuticals-17-00098-f004:**
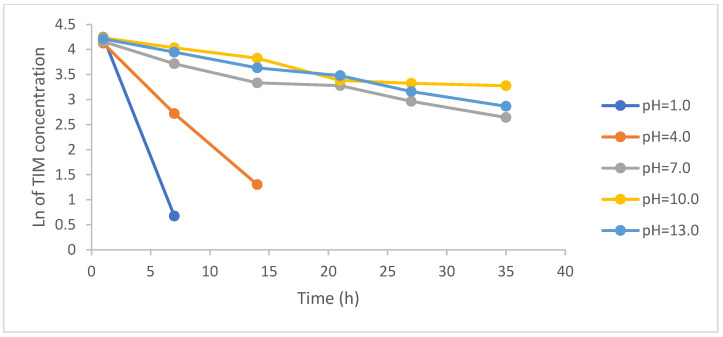
First-order plots of degradation of timolol (TIM) at different pH, under UV/VIS irradiation with energies 2700–94,510 kJ/m^2^ (1–35 h); the results are presented as mean for three repetitive measurements.

**Figure 5 pharmaceuticals-17-00098-f005:**
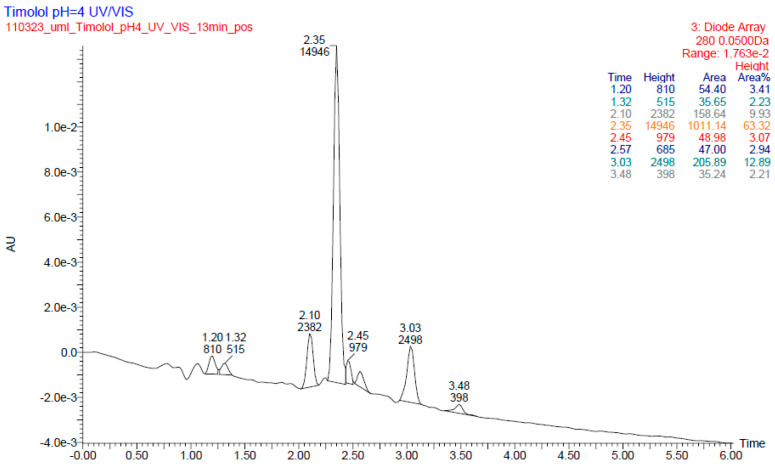
UPLC-UV chromatogram (at 280 nm) of timolol (TIM) with t_R_ = 3.03 min and its degradation products (t_R_ = 2.10, 2.35, 2.45, 2.57 and 3.48 min) after UV/VIS irradiation at pH 4.

**Figure 6 pharmaceuticals-17-00098-f006:**
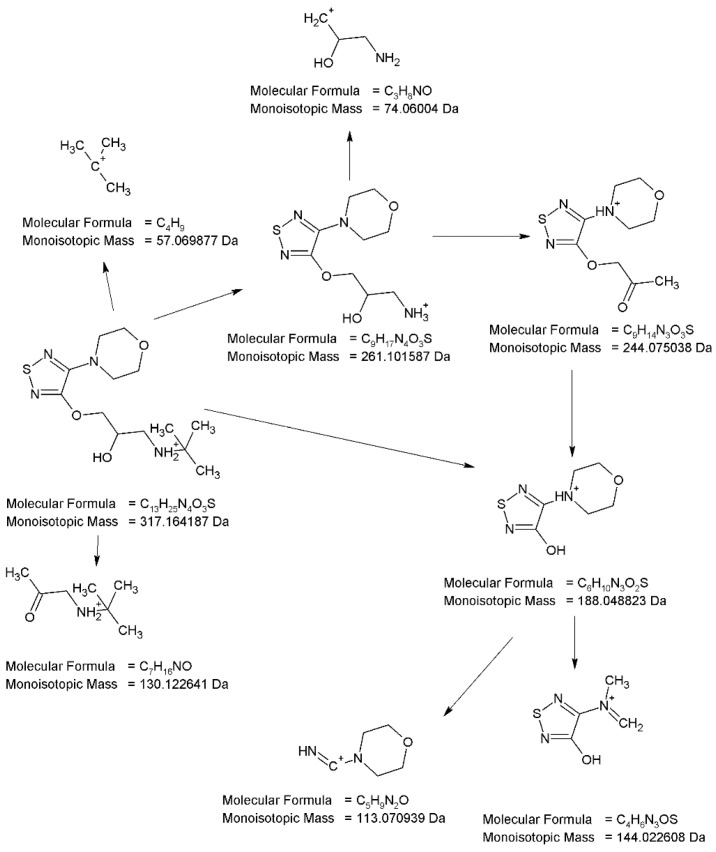
Proposed fragmentation pattern of timolol (TIM).

**Figure 7 pharmaceuticals-17-00098-f007:**
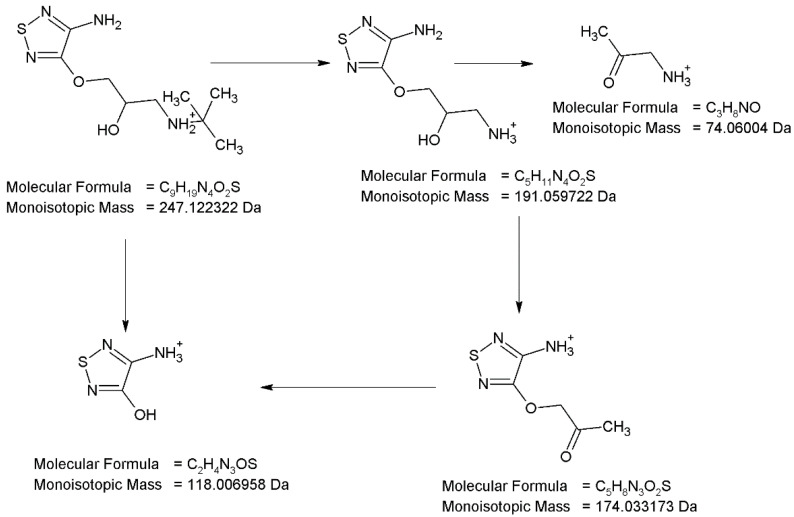
Proposed fragmentation pattern of degradation product PD1.

**Figure 8 pharmaceuticals-17-00098-f008:**
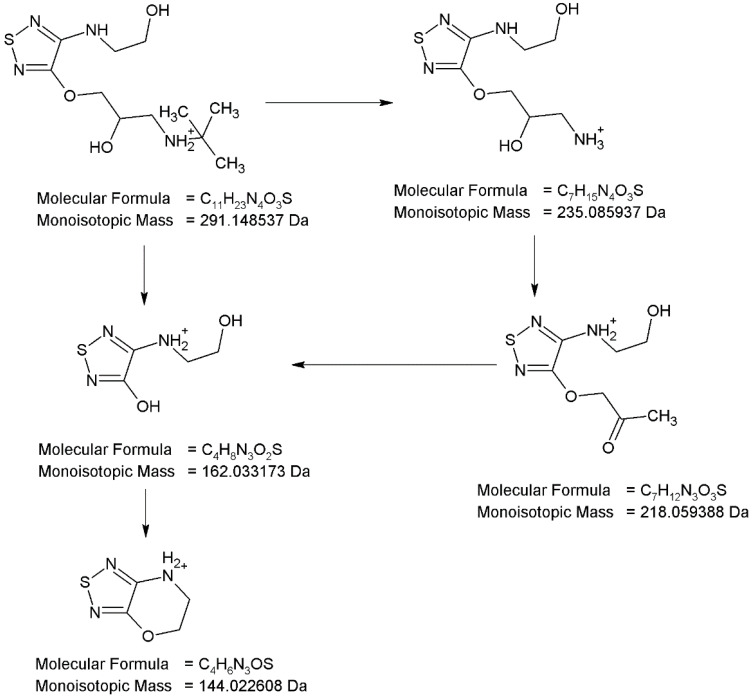
Proposed fragmentation pattern of degradation product PD2.

**Figure 9 pharmaceuticals-17-00098-f009:**
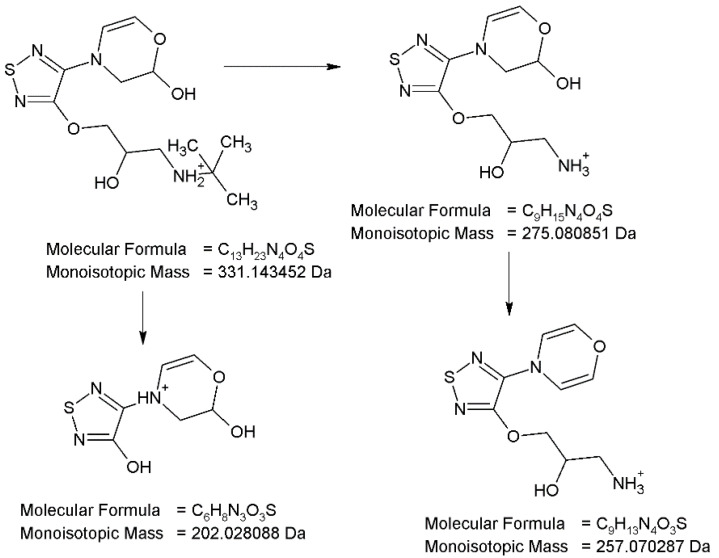
Proposed fragmentation pattern of degradation product PD3.

**Figure 10 pharmaceuticals-17-00098-f010:**
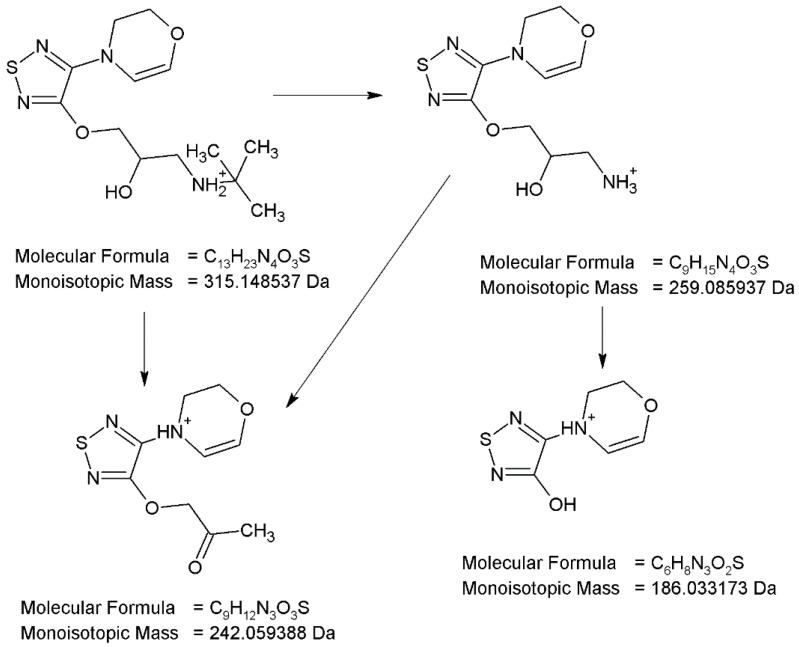
Proposed fragmentation pattern of degradation product PD4.

**Figure 11 pharmaceuticals-17-00098-f011:**
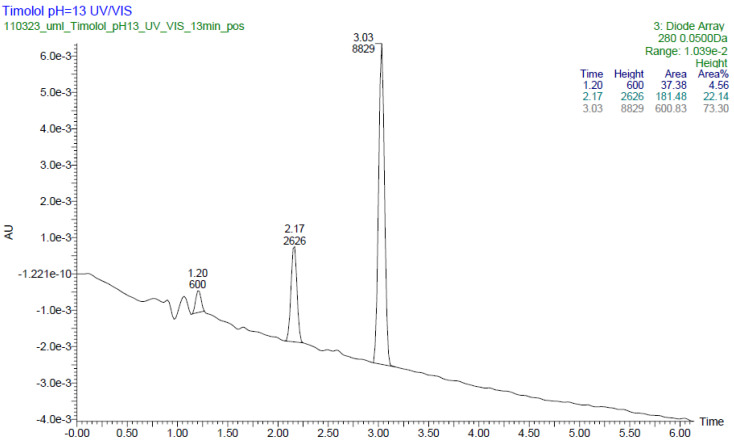
UPLC-UV chromatogram at 280 nm of timolol (TIM) with t_R_ = 3.03 min and its degradation product with t_R_ = 2.17 min after UV/VIS irradiation at pH 13.

**Figure 12 pharmaceuticals-17-00098-f012:**
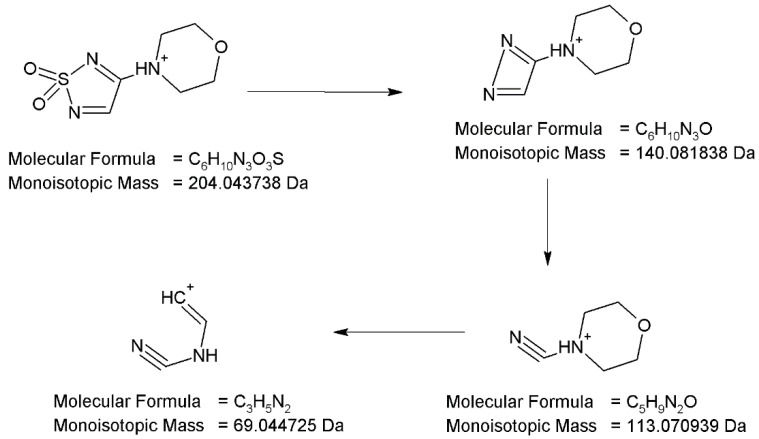
Proposed fragmentation pattern of degradation product PD5.

**Figure 13 pharmaceuticals-17-00098-f013:**
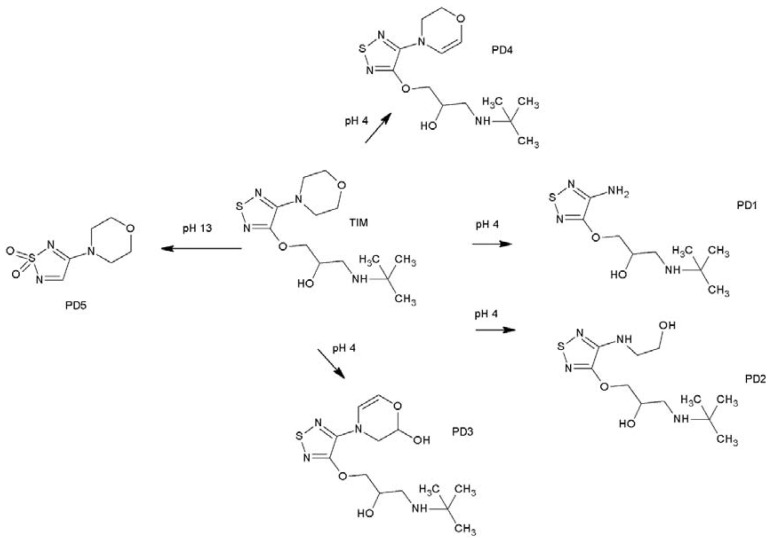
Proposed photodegradation products of timolol (TIM) at different pH conditions.

**Figure 14 pharmaceuticals-17-00098-f014:**
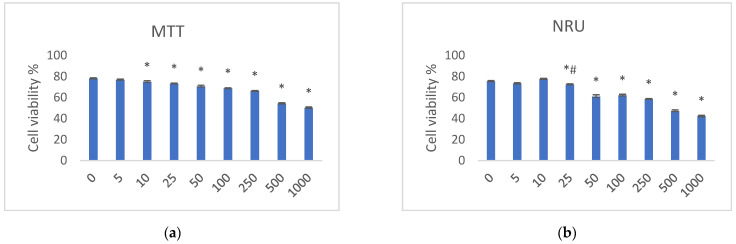
The results of MTT (**a**) and NRU (**b**) tests showing decreased cell viability after irradiation with 5 J/cm^2^ and 5–1000 µg/mL of timolol (TIM). The results are presented as mean ± SEM (standard error of the mean, n = 3). Statistical significance of differences between the cells irradiated in the presence of TIM (5–1000 µg/mL) and without TIM (0 µg/mL) was set at 0.005 (*) or 0.02 (#).

**Figure 15 pharmaceuticals-17-00098-f015:**
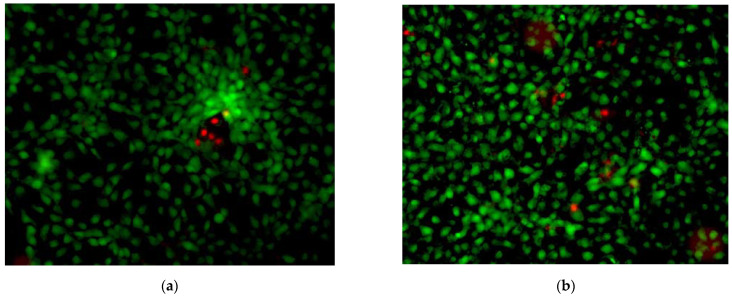
Representative photo obtained in Live/DEAD test with BALB/c 3T3 fibroblasts irradiated with UV/VIS light in the presence of timolol (TIM) at a concentration of 1000 µg/mL (**a**) or without TIM (**b**), using a fluorescence microscope with magnification 100×; the live cells with calcein (CA) are green, while the dead cells with ethidium homodimer-1 (EthD-1) are red.

**Table 1 pharmaceuticals-17-00098-t001:** Validation parameters of the quantitative LC-UV assay of timolol (TIM).

Parameter	Value
Linearity range	10–100 µg/mL
Regression equation (mean ± SD, n = 5)	y = 0.02322 ± 0.00023x + 0.03935 ± 0.00849
R^2^ (mean ± SD, n = 5)	0.9979 ± 0.0031
LOD	1.21 µg/mL
LOQ	3.66 µg/mL
Recovery (mean ± SD, n = 9)	99.62 ± 1.14%
RSD for within-day precision (n = 3)	1.07%
RSD for between-day precision (n = 9)	2.27%
RSD for the peak areas (n = 5)	1.17%

SD—standard deviation; RSD—relative standard deviation.

**Table 2 pharmaceuticals-17-00098-t002:** Photodegradation of timolol (TIM) in solutions of different pH.

pH	Degradation [%]	y = ax + b	R^2^	k [s^−1^]	t_0.1_ [h]	t_0.5_ [h]
1	98.13	-	-	-	-	-
4	96.29	-	-	-	-	-
7	86.05	y = −0.0416x + 4.0765	0.9663	2.7 × 10^−5^	1.08	7.13
10	73.85	y = −0.0309x + 4.2203	0.9248	1.9 × 10^−5^	1.54	10.13
13	82.38	y = −0.0389x + 4.2312	0.9663	2.5 × 10^−5^	1.17	7.70

**Table 3 pharmaceuticals-17-00098-t003:** Singlet oxygen (^1^O_2_) and superoxide radical anion (O_2_**^−^**) generation under UV/VIS irradiation of timolol (TIM) at pH 7.4.

	49 kJ/m^2^	234 kJ/m^2^	675 kJ/m^2^	1350 kJ/m^2^	2025 kJ/m^2^	2700 kJ/m^2^
^1^O_2_						
TIM	11	30	52	77	130	201
Quinine	98	156	282	319	401	493
Benzocaine	−13	−10	−9	−8	−3	10
O_2_^−^						
TIM	7	53	92	101	124	170
Quinine ^1^	81	127	133	148	179	221
Benzocaine ^2^	−13	−13	−11	−8	−6	2

^1^ positive control; ^2^ negative control.

## Data Availability

Data is contained within the article.

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
