# Peer review of "The Study on Timolol and Its Potential Phototoxicity Using Chemical, In Silico and In Vitro Methods"

_pharmaceuticals, 2024, doi:10.3390/ph17010098_

Round 1
Reviewer 1 Report
Comments and Suggestions for Authors
The study investigates the photodegradation and potential phototoxicity of Timolol (TIM), a non-selective ß-adrenergic receptor antagonist used in various clinical applications. The research employs a multi-faceted approach, combining chemical, in silico, and in vitro methods to comprehensively assess the risk factors associated with TIM.
Questions:
Timolol pharmacological profile should be discussed in the Introductio section.
Timolol stereochemistry should ve discussed in the Introduction section.
Timolol structure should be presented in Introduction section.
Based on the clinical applications of Timolol why is its photodegradation and phototoxicity important in these contexts?
At what concentrations of Timolol did the study observe a significant inhibition of cell viability, especially in the context of UV/VIS irradiation?
How might the potential phototoxicity of TIM impact its clinical use, and are there any suggested measures or precautions?
Were there any limitations acknowledged in the study, and do the findings suggest avenues for future research or clinical considerations?
Comments on the Quality of English LanguageEnglish is OK, minor corrections can be done
Author Response
Reviewer 1
The study investigates the photodegradation and potential phototoxicity of Timolol (TIM), a non-selective ß-adrenergic receptor antagonist used in various clinical applications. The research employs a multi-faceted approach, combining chemical, in silico, and in vitro methods to comprehensively assess the risk factors associated with TIM.
Questions:
Timolol pharmacological profile should be discussed in the Introduction section.
Timolol stereochemistry should ve discussed in the Introduction section.
Timolol structure should be presented in Introduction section.
Thank you for these suggestions. Some new sentences on timolol pharmacology as well as timolol structure and stereochemistry were added in the text (lines 37-48). Also, some new references were added.
Based on the clinical applications of Timolol why is its photodegradation and phototoxicity important in these contexts?
According to the literature, some incidents of photosensitization have been reported for beta-blockers used in the treatment of hypertension as well as arrhythmias and heart failure, for example bisoprolol and atenolol have been associated with photoinduced cutaneous reactions. As was mentioned in our paper, beta-blockers could also be effective in some dermatologic diseases as hemangiomas, wound healing, Kaposi sarcoma, melanoma, pyogenic granuloma, and pemphigus. For these reasons, it is very important to procure more data about their cutaneous adverse effects because toxic responses could be elicited by topically administered photoreactive drugs after the exposure of patients to environmental light.
Some additive sentences were added in the text (lines 41-47 and 235-238).
At what concentrations of Timolol did the study observe a significant inhibition of cell viability, especially in the context of UV/VIS irradiation?
According to MTT results, viability of fibroblasts irradiated in the presence of TIM was in the range of 78.21-50.22%, with increasing TIM concentration from 5 to 1000 µg/ml. A significant (at p < 0.005) decrease in cell viability was observed (in relation to the control group irradiated without TIM) at concentration of 10 µg/ml while the concentration at which the cell viability was reduced by approximately 50% was 500 µg/ml. Using NRU test, a visible drop in cell viability in the range 75.44-42.32% was observed with increasing TIM concentrations. Statistically significant impact of TIM was observed at concentration 25 µg/ml at p < 0.02, and at concentration 50 µg/ml at p < 0.05 (in relation to the control group irradiated without TIM). At the same time, irradiated fibroblasts with the concentration of TIM equal 500 µg/ml showed the reduced cell viability by approximately 50%.
How might the potential phototoxicity of TIM impact its clinical use, and are there any suggested measures or precautions?
We believe that summarizing the current knowledge on frequently used phototoxic medications can help clinicians and other professionals stay informed of the known and possible unknown risks of phototoxicity and promote better patient education by providing clear photoprotective instructions, avoidance of ultraviolet light while taking phototoxic medications, and advising patients on the short and long-term risks to minimize adverse effects.
Were there any limitations acknowledged in the study, and do the findings suggest avenues for future research or clinical considerations?
Some limitations of the present study concern reliability of the methods used for estimation, and sometimes possibility of overestimation or underestimation of the risk of the drug phototoxicity. Main limitations of such studies, however, are those that they are not able to assess the impact of metabolic modifications of drugs in patients, as well as their individual modifications due to genetic differences. Some additive sentences in this area were added in our conclusions (lines 426-430).
Reviewer 2 Report
Comments and Suggestions for Authors
Line 58 change was to were
Line 64 change to Results and Discussion
Line 72 The molar extinction coefficient should be given as: M-1cm-1
Line 82 the full names of the acronyms LOD and LOQ should be given here ( the first time mentioned in the text) and not further below in lines 295 and 296 respectively.
Sentence in Lines 108-110 needs rephrasing. In line 109 ...the both... shlould be deleted
Table 2 page 5 insert space before and after = (i.e. y = -value
The sentence in lines 228-229 cannot make sense and should be rephrased.
Line 236 give the full name of the acronym MMT.
Line 244 give the full name of the acronym NRU.
Line 257 What is compound CA-AM? give a full name.
Line 299 give the full name of the acronym RSD
Line 364 give the full name of the acronym HBSS
Line 393 insert spaces p ≤ 0.05
Comments on the Quality of English Language
Minor corrections need as pointed out to the authors.
Author Response
Reviewer 2
Thank you very much for your positive opinion about our article and for all your comments.
Line 58 change was to were
Corrected
Line 64 change to Results and Discussion
Corrected
Line 72 The molar extinction coefficient should be given as: M-1cm-1
Corrected
Line 82 the full names of the acronyms LOD and LOQ should be given here ( the first time mentioned in the text) and not further below in lines 295 and 296 respectively.
Corrected
Sentence in Lines 108-110 needs rephrasing. In line 109 ...the both... should be deleted
Corrected
Table 2 page 5 insert space before and after = (i.e. y = -value)
Corrected
The sentence in lines 228-229 cannot make sense and should be rephrased.
Corrected
Line 236 give the full name of the acronym MTT.
Corrected (line 245)
Line 244 give the full name of the acronym NRU.
Corrected (line 237)
Line 257 What is compound CA-AM? give a full name.
Corrected (line 266)
Line 299 give the full name of the acronym RSD
Corrected
Line 364 give the full name of the acronym HBSS
Corrected (line 361)
Line 393 insert spaces p ≤ 0.05
Corrected
Reviewer 3 Report
Comments and Suggestions for Authors
The manuscript describes the assessment of the of possible photocytotoxicity of timolol, a beta blocker drug. Five possible photodegradation products have been identified and their structures have been proposed based on high-resolution mass spectra analysis and fragmentation patterns. In vitro tests were applied to examine the effect of the drug on fibroblasts after UV/Vis irradiation. The aim of the research is clear, and the results may be useful for safer use of the drug, especially given that only a few similar studies have been conducted so far.
The manuscript is well organized and well written; however, I would suggest a revision of the title. The current title begins with a question, but there is no question mark, and in any case, it would be better to highlight specific insights.
Throughout the manuscript, 1O2 and O2×- should be used for singlet oxygen and superoxide radical anion, respectively, instead of SO and SA.
The weakest part of the manuscript is the detection of reactive oxygen species, in the Results part (2.5. ROS Generation) and in the Materials and Methods section (3.8. ROS Assays). It is not at all clear how ROS was measured (reference 14 cannot be checked, or it has not been cited properly), so the details of the measurement should be given, and the results should be presented and discussed accordingly, in particular the data in Table 4.
Author Response
Reviewer 3
The manuscript describes the assessment of the of possible photocytotoxicity of timolol, a beta blocker drug. Five possible photodegradation products have been identified and their structures have been proposed based on high-resolution mass spectra analysis and fragmentation patterns. In vitro tests were applied to examine the effect of the drug on fibroblasts after UV/Vis irradiation. The aim of the research is clear, and the results may be useful for safer use of the drug, especially given that only a few similar studies have been conducted so far.
The manuscript is well organized and well written; however, I would suggest a revision of the title. The current title begins with a question, but there is no question mark, and in any case, it would be better to highlight specific insights.
Thank you very much for this positive opinion about our article and for all your comments. The title was changed according to the above suggestion.
Throughout the manuscript, 1O2 and O2×- should be used for singlet oxygen and superoxide radical anion, respectively, instead of SO and SA.
Corrected throughout the text and tables.
The weakest part of the manuscript is the detection of reactive oxygen species, in the Results part (2.5. ROS Generation) and in the Materials and Methods section (3.8. ROS Assays). It is not at all clear how ROS was measured (reference 14 cannot be checked, or it has not been cited properly), so the details of the measurement should be given, and the results should be presented and discussed accordingly, in particular the data in Table 4.
Thank you once more for this important suggestion. Respective parts are corrected including all important data (lines 224-238). Further, the doi for the Ref. 14 (now Ref. 17) was added to make it easier to find this item.
Round 2
Reviewer 1 Report
Comments and Suggestions for Authors
S and R - timolol should be written in italic
Figure 2 is not understandable, please inserted in colours
Figure 3 seem to have a background colour
The gradations on the axis should be less frequent in Figure 2/3
Chromatogram should be more carefully checked
The manuscript doesn't respect entirely Journal recommandation regarding format
Comments on the Quality of English LanguageEnglish is OK, there are some typo and syntax errors in the text
Author Response
Reviewer 1
S and R - timolol should be written in italic
Corrected.
Figure 2 is not understandable, please inserted in colours
Corrected. The spectrum before irradiation is black while the one after UV/VIS irradiation is red.
Figure 3 seem to have a background colour
Corrected, the background colour is deleted.
The gradations on the axis should be less frequent in Figure 2/3
Thank you for this suggestion. However, these drawings are original prints from the spectrophotometer and chromatograph, respectively. Therefore, it was decided to keep the proposed gradations.
Chromatogram should be more carefully checked
Thank you for this suggestion. Our chromatograms were checked and corrected.
The manuscript doesn't respect entirely Journal recommandation regarding format.
Thank you for this suggestion. The manuscript was corrected according to the format of Pharmaceuticals.
English is OK, there are some typo and syntax errors in the text.
Some English errors were found and corrected.
Reviewer 3 Report
Comments and Suggestions for Authors
The authors have responded to the remarks and questions raised in the review and improved the manuscript; I suggest accepting it for publication.
Author Response
Reviewer 3
The authors have responded to the remarks and questions raised in the review and improved the manuscript; I suggest accepting it for publication.
Thank you very much for your comment.